



# Quantitative evaluation of numerical integration schemes for Lagrangian particle dispersion models

Huda Mohd. Ramli[1] and J. Gavin Esler[1]

[1]Department of Mathematics, University College London, London, UK

*Correspondence to:* J. Gavin Esler
(j.g.esler@ucl.ac.uk)

**Abstract.**

A rigorous methodology for the evaluation of integration schemes for Lagrangian particle dispersion models (LPDMs) is presented. A series of one-dimensional test problems are introduced, for which the Fokker-Planck equation is solved numerically using a finite-difference discretisation in physical space, and a Hermite function expansion in velocity space. Numerical convergence errors in the Fokker-Planck equation solutions are shown to be much less than the statistical error associated with a practical-sized ensemble ($N = 10^6$) of LPDM solutions, hence the former can be used to validate the latter. The test problems are then used to evaluate commonly used LPDM integration schemes. The results allow for optimal time-step selection for each scheme, given a required level of accuracy. The following recommendations are made for use in operational models. First, if computational constraints require the use of moderate to long time-steps it is more accurate to solve the random displacement model approximation to the LPDM, rather than use existing schemes designed for long time-steps. Second, useful gains in numerical accuracy can be obtained, at moderate additional computational cost, by using the relatively simple 'small-noise' scheme of Honeycutt.

## 1 Introduction

State-of-the-art Lagrangian particle dispersion models (LPDMs hereafter), for example FLEXPART (Stohl et al., 2005) and NAME (Jones et al., 2007), are key scientific tools for the study of the long-range transport and dispersal of the transport of atmospheric trace gases and aerosols. Applications are diverse, e.g. establishing the relationship between emissions of pollutants and air quality downstream (Cassiani et al., 2012), aerosol dispersal following volcanic eruptions (Devenish et al., 2011; D'Amours et al., 2010), modelling of nuclear accident scenarios (Stohl et al., 2012), and determination of constraints on chemical emissions via inverse modelling (Seibert and Frank, 2004; Stohl et al., 2010). More fundamentally, LPDMs can be used to address key scientific questions concerning the nature of transport in the atmosphere (Legras et al., 2005; Berthet et al., 2007), including how transport might be influenced by a changing climate.

Mathematically speaking, LPDMs are formulated as stochastic differential equations (SDEs hereafter). (It is notable that it is possible to include jump processes (Platen and Liberati, 2010) as a representation of non-local convective parametrisations (Forster et al., 2007), but we will not be concerned here with this possibility.) Although the numerical analysis of



solution techniques for SDEs (e.g. Kloeden and Platen, 1992; Milstein and Tretyakov, 2004) is a mature subject in mathematics, LPDMs have not, generally speaking, exploited developments in the subject, and are typically formulated using numerical schemes adapted from those used for ordinary differential equations (see e.g. Stohl et al., 2005). Validation of LPDMs has focussed instead on direct comparison with observational data (Stohl et al., 1998; Ryall and Maryon, 1998). Our contention is

that observational comparison, while clearly a necessary aspect to model development, will be insufficient if any uncertainty exists concerning the accuracy of the numerical solution of the underlying equations.

The aim of the present work, therefore, is to introduce a rigorous framework for the testing and evaluation of numerical schemes for LPDMs. The framework is based on a standard one-dimensional dispersion model problem (Rodean, 1996; Wilson and Sawford, 1996) modelling the vertical dispersion of air parcels in the atmospheric boundary layer (ABL hereafter). Vertical

profiles of turbulent statistics representative of both stable and neutral conditions will be considered, and the LPDM equations will be of the 'well-mixed' class (Thomson, 1987), meaning that long time probability distribution of the solutions (the invariant measure of the SDEs) is given by a pre-specified 'atmospheric' distribution (taken here to be uniform in physical space and Gaussian in velocity-space). Hence the model problem, while idealised, captures key elements of the physics of dispersion in the stable and neutral ABL. The convective case, in which the vertical velocity statistics are non-Gaussian (see e.g. Cassiani

et al., 2015), will require a separate test case and is not discussed here.

Our approach to evaluating a given LPDM numerical scheme is to cross-validate its performance against a numerical solution of the corresponding Fokker-Planck equation (FPE hereafter, see e.g. Gardiner, 2009). The FPE describes the time-evolution of the probability density function (pdf) of the stochastic process, and is formulated in position-velocity space, so in the context of the current problem of dispersion in one spatial dimension, is a partial differential equation in 2+1 dimensions. Note that in three

spatial dimensions in which the FPE is a 6+1 dimensional PDE, it will be computationally impractical in most circumstances to obtain accurate solutions to the FPE, and consequently LPDMs will be the only practical tool to solve the problem.

A solution method based on a Hermite function expansion is introduced in order to obtain accurate solutions of the FPE with computational efficiency. Evaluation of the LPDM scheme proceeds by a comparison of pdfs in appropriate error norm, where the LPDM pdf is generated from an ensemble of solutions, using the kernel density method (e.g. Silverman, 1986; Wand

and Jones, 1994). The performance of various schemes are evaluated, as a function of time-step $\Delta t$, including the textbook (basic) Euler-Maruyama scheme, the second-order and third-order weak Runge-Kutta scheme of Platen (see §15.1 of Kloeden and Platen, 1992), the 'small-noise' second-order Runge-Kutta method of Honeycutt (Honeycutt, 1992), the 'long time-step' scheme used operationally in FLEXPART (Stohl et al., 2005) and a suggested improvement to this last scheme.

The outline of the work is as follows. In section 2, the SDEs describing the evolution of particle trajectories in the LPDM

are introduced, together with the corresponding FPE. A numerical solution scheme for the FPE is described and solutions are obtained and benchmarked for a number of test cases. In section 3, the methodology for using the FPE solution to assess specific numerical schemes for the LPDM is presented, and in section 4 this methodology is then applied to specific schemes discussed above. In section 5 the consequences of our findings are discussed and conclusions are drawn.



## 2 The model problem

### 2.1 The model problem formulated as an LPDM

Consider a horizontally homogeneous turbulent ABL of uniform density, with a vertical velocity distribution that is Gaussian with zero mean and standard deviation $\sigma_w(z)$, and which has Lagrangian decorrelation time-scale $\tau(z)$. The canonical stochastic differential equation model (e.g. Rodean, 1996; Wilson and Sawford, 1996) for one-dimensional vertical dispersion in the ABL is

$$\mathrm{d}W_t = \left( -\frac{W_t}{\tau} + \frac{1}{2}\left(1 + \left(\frac{W_t}{\sigma_w}\right)^2\right)\frac{\partial \sigma_w^2}{\partial z} \right)\mathrm{d}t + \left(\frac{2\sigma_w^2}{\tau}\right)^{1/2}\mathrm{d}B_t, \tag{1}$$

$$\mathrm{d}Z_t = W_t\,\mathrm{d}t.$$

Here $W_t$ and $Z_t$ are the vertical velocity and height of a given air parcel. Both are stochastic variables, with each individual realisation determined by that of the Brownian (or Wiener) process $B_t$. Further $\sigma_w = \sigma_w(Z_t)$ and $\tau = \tau(Z_t)$ are the values of $\sigma_w(z)$ and $\tau(z)$ local to the parcel. In operational LPDMs, such as FLEXPART, appropriate vertical profiles for $\sigma_w(z)$ and $\tau(z)$ are specified based on empirical fits to observations of different ABL conditions, as will be discussed below. The equation set (1) is typically augmented with reflecting boundary conditions at the Earth's surface and at the ABL top (see Thomson et al., 1997, for detailed discussion of the top boundary condition). For definiteness, for our test case runs, the initial velocity for (1) at $t = 0$ is sampled from a normal distribution $W_0 \sim \mathcal{N}(0, \sigma_w^2(z_0))$ and, for ease of comparison to the FPE results below, the initial position is sampled from a distribution $Z_0 \sim \mathcal{N}(z_0, \sigma_z^2)$ centred on an initial height $z_0$ with standard deviation $\sigma_z$.

For the purposes of numerical solution, it is more convenient (e.g. sec. 3.1 of Rodean, 1996) to use Ito's lemma to express (1) in terms of the variables $\Omega_t = W_t/\sigma_w(Z_t)$ and $Z_t$, leading to

$$\mathrm{d}\Omega_t = \left( -\frac{\Omega_t}{\tau} + \frac{\partial \sigma_w}{\partial z} \right)\mathrm{d}t + \left(\frac{2}{\tau}\right)^{1/2}\mathrm{d}B_t, \qquad\qquad \Omega_0 \sim \mathcal{N}(0,1) \tag{2}$$

$$\mathrm{d}Z_t = \Omega_t \sigma_w\,\mathrm{d}t, \qquad\qquad Z_0 \sim \mathcal{N}(z_0, \sigma_z^2).$$

The simpler form (2) is exactly equivalent to (1). Moreover, the FPE of (2) has a considerably simpler form than the corresponding FPE of (1), a fact which will prove useful below.

It is simplest to view equation (2) as a non-dimensional equation, given that in particular $\Omega_t$ is already a non-dimensional variable. The natural non-dimensionalisation has length, velocity and time scales of ABL height $h$, surface friction velocity $u_*$ and $h/u_*$ respectively. Under this non-dimensionalisation, the spatial domain for (2) is $0 \leq Z_t \leq 1$.

To specify our test-case problems it is necessary to choose suitable (non-dimensional) profiles for $\sigma_w(z)$ and $\tau(z)$. Here we choose to focus on three such profiles, two of which are widely used (Hanna, 1982; Stohl et al., 2005) empirical fits to observed statistics in stable and neutral conditions respectively. The third has $\tau(z)$ constant and a linear profile for $\sigma_w(z)$, and is used to demonstrate a new LPDM scheme introduced below. The details of the profiles used are given in Table 1. In practice, the exact profiles suggested by Hanna (1982) are modified slightly, as detailed in Table 1, to avoid singular behaviour at the ABL



top and bottom. This is necessary because in Hanna's original profiles either $\sigma_w \to 0$ or $\tau \to 0$ as $z \to 0, 1$ with neither type of behaviour being physical.

In section 4 large ensembles of numerical solutions of equation (2) will be calculated using different numerical integration schemes. The accuracy of each numerical scheme, as a function of time-step $\Delta t$, will be assessed by comparison with the

corresponding solution of the FPE, to be detailed next.

### 2.2 The model problem formulated as an FPE

Following the standard procedure in stochastic calculus, (e.g. sec. 3.4.1 of Gardiner, 2009), the FPE which describes the the time-evolution of the probability density $p(\omega, z, t)$ of $(\Omega_t, Z_t)$ in (2), can be obtained as

$$\frac{\partial p}{\partial t} = -\frac{\partial (\omega \sigma_w p)}{\partial z} - \frac{\partial}{\partial \omega}\left(\left(-\frac{\omega}{\tau} + \frac{\partial \sigma_w}{\partial z}\right)p\right) + \frac{1}{\tau}\frac{\partial^2 p}{\partial \omega^2}. \tag{3}$$

The initial conditions consistent with those given in (2) are (for $\sigma_z \ll 1$ and $z_0$ not near the boundaries)

$$p(\omega, z, 0) = \frac{1}{2\pi\sigma_z}\exp\left(-\frac{\omega^2}{2} - \frac{(z - z_0)^2}{2\sigma_z^2}\right). \tag{4}$$

The FPE (3) also requires boundary conditions at $z = 0, 1$ which are consistent with the reflecting boundary conditions for the LPDM. The boundary conditions consistent with reflection are

$$p(\omega, 0, t) = p(-\omega, 0, t), \quad p(\omega, 1, t) = p(-\omega, 1, t). \tag{5}$$

which in probabilistic terms is equivalent to the reflection condition $\Omega_t \to -\Omega_t$ being applied at the boundaries. Wilson et al. (1993) found that this perfect reflection algorithm is exactly consistent with the 'well-mixed constraint' in homogeneous Gaussian turbulence (see also appendix of §11 of Rodean, 1996).

Equations (3-5) constitute a well-defined initial-value problem which is suitable for numerical solution. An important quantity obtained from the solution $p(\omega, z, t)$ is the physical concentration of parcels given by

$$c(z, t) = \int_{-\infty}^{\infty} p(\omega, z, t)\,\mathrm{d}\omega. \tag{6}$$

The concentration $c(z, t)$ will be our main benchmark quantity in section 4 below.

### 3 Numerical solution of the FPE

### 3.1 The Hermite expansion for the FPE

The non-dimensionalised FPE (3) is a hypo-elliptic differential equation defined on $\mathbb{R} \times [0, 1]$. Our approach to its numerical

solution is to seek a solution based on the following Hermite polynomial expansion

$$p(\omega, z, t) = \frac{1}{\sqrt{2\pi}}\sum_{k=0}^{\infty} C_k(z, t)\,\mathrm{He}_k(\omega)\,\mathrm{e}^{-\omega^2/2}. \tag{7}$$





Here the functions $C_k(z,t)$ denote the projection, at the vertical level and time $(z,t)$, of $p(\omega,z,t)$ onto the (probabilists') Hermite function $\mathrm{He}_k(\omega)\mathrm{e}^{-\omega^2/2}/\sqrt{2\pi}$ where $\mathrm{He}_k(\omega)$ is the Hermite polynomial defined by

$$\mathrm{He}_k(\omega) = (-1)^k \, \mathrm{e}^{\omega^2/2} \frac{\mathrm{d}^k}{\mathrm{d}\omega^k} \mathrm{e}^{-\omega^2/2}. \tag{8}$$

Notice that it follows that the particle concentration (6) satisfies $c(z,t) = C_0(z,t)$.

Before inserting the expansion (7) into the FPE (3) it is helpful to rewrite the FPE in the form

$$\frac{\partial p}{\partial t} = \frac{1}{\tau}\left(\frac{\partial^2 p}{\partial \omega^2} + \omega\frac{\partial p}{\partial \omega} + p\right) - \frac{\partial}{\partial \omega}\left(\frac{\partial \sigma_w}{\partial z}p\right) - \frac{\partial(\omega\sigma_w p)}{\partial z}. \tag{9}$$

In this form the Hermite function identity (A2) can be used to evaluate the first term on the right-hand side. Further, the second and third terms on the right-hand side can be simplified using the derivative and recursion formulae for Hermite polynomials (A4), (A5). After some working the result is (using the convention $C_{-1} \equiv 0$)

$$\sum_{k=0}^{\infty}\mathrm{He}_k(\omega)\mathrm{e}^{-\omega^2/2}\left(\frac{\partial C_k}{\partial t} + \frac{k}{\tau}C_k + (k+1)\frac{\partial}{\partial z}(\sigma_w C_{k+1}) + \sigma_w\frac{\partial C_{k-1}}{\partial z}\right) = 0. \tag{10}$$

Using the orthogonality property of Hermite functions (A3) it follows that

$$\frac{\partial C_0}{\partial t} = -\frac{\partial}{\partial z}(\sigma_w C_1)$$
$$\frac{\partial C_k}{\partial t} = -\frac{k}{\tau}C_k - (k+1)\frac{\partial}{\partial z}(\sigma_w C_{k+1}) - \sigma_w\frac{\partial C_{k-1}}{\partial z}, \qquad \text{for } k \geq 1. \tag{11}$$

The system (11) constitutes an infinite system of coupled 1+1 dimensional first-order partial differential equations for the

coefficients $C_k$. For a numerical solution this series can be truncated as we describe below.

The initial conditions for (11) are easily obtained from (4) using the orthogonality property. The boundary conditions can be obtained using the symmetry $\mathrm{He}_k(\omega) = (-1)^k\,\mathrm{He}_k(-\omega)$. Substituting the expansion (7) into the boundary condition (5), it follows that

$$\sum_{k \text{ odd}} C_k(z,t)\mathrm{He}_k(\omega)\frac{\mathrm{e}^{-\omega^2/2}}{\sqrt{2\pi}} = 0, \text{ at } z = 0, 1. \tag{12}$$

and consequently

$$C_k(0,t) = C_k(1,t) = 0, \quad \text{for } k \text{ odd}. \tag{13}$$

It may seem surprising that the even equations have no boundary condition and the odd equations take two boundary conditions. However, as the system (11) consists of *first-order* PDEs it is clear that the total number of boundary conditions will be correct, provided that the series is truncated at $k = K$ odd.

It is worth noting that the series (11) can also be truncated at $K = 0$ by using an (approximate) quasi-stationary balance in the $k = 1$ equation of the form

$$C_1 = -\sigma_w \tau \frac{\partial C_0}{\partial z}, \tag{14}$$





which results in the diffusion equation

$$\frac{\partial C_0}{\partial t} = \frac{\partial}{\partial z}\left(\sigma_w^2 \tau \frac{\partial C_0}{\partial z}\right), \quad \frac{\partial C_0}{\partial z}(0,t) = \frac{\partial C_0}{\partial z}(1,t) = 0. \tag{15}$$

It is well-known (e.g. section 3.5 of Thomson, 1987) that the LPDM (1) can be approximated by a random walk ('random displacement' or RDM) model

$$5 \quad \mathrm{d}Z_t = \frac{\partial}{\partial z}\left(\sigma_w^2 \tau\right)\mathrm{d}t + \left(2\sigma_w^2 \tau\right)^{1/2}\mathrm{d}B_t \tag{16}$$

Equation (15) is simply the Fokker-Planck equation of the RDM model (16), with the diffusivity $\kappa$ of the RDM being $\kappa = \sigma_w^2 \tau$. It is much easier to obtain accurate solutions of (16), compared to (1) at relatively large time-steps, hence an interesting question concerns when exactly it is preferable to solve (15) rather than (1). This question is best answered by quantifying the difference between the solution of (15) and (11) and using this difference as a benchmark for assessing the errors in LPDM calculations, as will be done in section 4 below.

### 3.2 The numerical method and benchmark solutions for the FPE

Based on the analysis above, (3) can be solved numerically by integrating the system (11) with boundary conditions (13), truncated at $k = K$ odd. Our approach is to use a standard finite-difference discretisation with $N_z$ grid points, equally spaced with $\Delta z = 1/N_z$, on a staggered cell-centred grid (i.e. $z_i = (i - 1/2)\Delta z$, for $i = 1, \ldots, N_z$) in order to apply the boundary conditions at $z = 0, 1$ systematically. The details of the implementation of the boundary conditions are described in Appendix B.

The set (11) are stiff and a naive solution method would have the time-step $\Delta t$ bounded above by $\Delta t \lesssim \mathrm{Min}_z \tau(z)/K$, i.e. the timescale of exponential decay of the highest Hermite function mode. However, considerably longer time-steps can be used if an exponential time-stepping scheme is chosen. Our choice is the 'Exponential Time-Differencing fourth-order Runge-Kutta' (ETDRK4) scheme of Kassam and Trefethen (2005), with the 'linear' operator in that scheme taken to be first term on the right-hand side of (11) only, because it is this first term that is responsible for the stiffness of (11).

To obtain our benchmark solutions of (11) and therefore (3), tests of the convergence of the solutions as both $\Delta t$ and $\Delta z$ are decreased and $K$ is increased, have been performed. For all three case studies, it was found to be adequate to take $K = 19$ to obtain fully converged solutions, because the Hermite series was found to converge rapidly i.e. $|C_{19}| \lesssim 10^{-16}$ everywhere in the domain. Comparison of a sequence of solutions with $\Delta z = 1/N_z$ with $N_z = 2^7, 2^8, \ldots, 2^{12}$ revealed quadratic convergence with $\Delta z$ as expected for our scheme. Fig. 2 shows the relative error $\mathcal{E}_j(t)$, with reference to the next-highest resolution solution, in the $L_2$-norm for the mean concentration $c(z,t)$ at fixed times, for the two test cases. That is,

$$\mathcal{E}_j(t) = \left(\int_0^1 \left(C_0^j(z,t) - C_0^{j+1}(z,t)\right)^2 \mathrm{d}z\right)^{1/2} \tag{17}$$

where $C_0^j(z,t)$ denotes the solution with $N_z = 2^j$. Quadratic convergence is evident from the slope of the graphs in Fig. 2. For example, typical numerical errors at $N_z = 2^{12}$ (highest resolution) are $\mathcal{E}_{12}(t_1) = 9.7 \times 10^{-5}$ (stable boundary layer at $t_1 = 1$) and $1.3 \times 10^{-4}$ (neutral boundary layer at $t_1 = 3$) respectively. The numerical accuracy above is sufficient for benchmarking



our LPDM solutions, because the statistical error associated with reasonable-sized ensembles ($N = 10^6$) of the LPDM is of order $\mathcal{E}(t_1) \approx 10^{-2}$, as will be discussed below.

Fig. 3 shows snapshots of the particle concentration $c(z,t)$ for each of the three FPE benchmark solutions described above. The left panel shows the constant $\tau$ case, middle panel shows the stable ABL case and the right panel the neutral ABL. In all three cases particles are initialised close to $z = z_0 = 1/2$ and disperse to become well-mixed throughout the ABL at late times. The neutral and stable cases differ in that mixing is rather more rapid (in terms of the dimensional timescale $h/u_*$) for the stable case compared to the neutral case. Also, in the neutral case mixing is relatively slow towards the top the ABL where the amplitude of turbulent fluctuations decays exponentially.

## 4    Evaluation of numerical schemes for LPDMs

In this section, a range of textbook, commonly-used and new numerical schemes for LPDMs will first be introduced, and then evaluated using the FPE solutions described above as a benchmark. The task is somewhat simplified because the equation set (2) is time-independent (autonomous). Note that it may be necessary to modify some of the schemes described below if an ABL with time-dependent statistics is to be modelled with the same formal accuracy. Note that, in the terminology of SDE numerical schemes (Kloeden and Platen, 1992), we are able to use 'weak' schemes (convergent in probability) in addition to 'strong' schemes (convergent in path), because we are primarily interested in the concentration of particles, which can be obtained from the pdf $p(\omega, z, t)$. The rate of convergence of a scheme, as measured by quantities which depend on the pdf such as the concentration $c(z,t)$, with respect to the time-step $\Delta t$ is known as its 'weak' order (see e.g. chapter 9 of Kloeden and Platen, 1992). The weak order is the relevant measure of comparison between schemes for our study, and should not be confused with the 'strong' order of a scheme, which refers to the rate of convergence of solution paths with respect to specific stochastic realisations.

It is important to note, however, that it is by no means obvious that a given scheme will attain its formal weak order when solving (2). This is because the assumptions under which the weak order of each scheme is derived are not met in the case of (2) because of the reflection boundary conditions. It is therefore necessary to solve (2) explicitly to assess each scheme.

### 4.1    LPDM numerical schemes

Tables 2-3 summarise the SDE numerical schemes to be investigated. The first, most obvious scheme to test is the Euler-Maruyama (E-M) scheme (Maruyama, 1955), i.e. the simplest and lowest order time-stepping scheme for SDEs. Next, as with ordinary differential equations (ODEs), it is possible to construct schemes with higher orders of formal accuracy in the spirit of Runge-Kutta schemes for ODEs. Here we test the performance of Platen's 'explicit order 2.0 weak scheme' (EXPLICIT 2.0) and 'explicit order 3.0 weak scheme' (EXPLICIT 3.0) (see chapter 15 of Kloeden and Platen, 1992). In common with schemes for ODEs, higher order schemes are somewhat more complicated to implement, and are more computationally expensive per time-step $\Delta t$. The advantage, however, is that the schemes have weak order $\Delta t^2$ (EXPLICIT 2.0) and $\Delta t^3$ (EXPLICIT 3.0) compared to $\Delta t$ for E-M.





A single candidate from a second class of schemes, the so-called 'small noise' schemes, to be investigated is the HON-SRKII scheme of Honeycutt (1992). Small noise schemes typically have the same weak order ($\Delta t$) as E-M (see e.g. discussion in Ch. 3 of Milstein and Tretyakov, 2004), but the schemes are designed so that the leading-order error depends on the 'noise amplitude' in the equation, which in many practical situations is sufficiently small that higher-order convergence is observed

in practice (at least for intermediate length time-steps, see discussion below). The HON-SRKII scheme will be shown below to converge with global error $\sim \Delta t^2$ in this intermediate time-step regime.

A third class of schemes to be investigated are designed to work well with long time-steps. Such schemes are of interest operationally, because the practical advantages of calculating large-ensembles efficiently are thought to outweigh the disadvantage of loss of accuracy due to time-stepping errors. The model FLEXPART (Stohl et al., 2005), for example, switches using

between E-M and a long time-stepping scheme due to Legg and Raupach (1982, LEGGRAUP). It is of some interest to verify that long time-stepping schemes such as LEGGRAUP do indeed outperform E-M at operationally relevant values of $\Delta t$. In fact, in Appendix C we review the derivation of the LEGGRAUP scheme, and show that additional care is needed to couple the velocity and position equations. A corrected scheme (LONGSTEP) is derived in Appendix C and is then compared with the schemes listed above in section 4.2.

The method used to compare the results from a particular scheme, at fixed time-step $\Delta t$, to the particle concentration $c(z, t)$ obtained from the numerical solution of the FPE, is as follows. First, a large-ensemble (typically $N = 10^6$) of trajectories is calculated using the scheme under investigation. Next, the density of particles $\widehat{c}$ is reconstructed from the resulting ensemble $\{Z_t^{(i)}, i = 1, ..., N\}$ using kernel density estimation,

$$\widehat{c}(z, t; h) = \frac{1}{Nh} \sum_{i=1}^{N} K\left(\frac{z - Z_t^{(i)}}{h}\right) + \text{'image terms'}. \tag{18}$$

Here $h > 0$ is a (small) smoothing parameter known as the bandwidth, and 'image terms' refer to contributions from the images of trajectories, introduced to satisfy the boundary conditions. The function $K(\cdot)$ is the kernel function, and is non-negative with zero mean and has unit integral. Here we use a Gaussian kernel. Details, including how the optimal bandwidth $h = h_*$ is chosen in practice, are given in Appendix D.

The error associated with a given scheme, at time-step $\Delta t$, is measured by the $L_2$-norm

$$\|c - \widehat{c}\|_2 = \left(\int_0^1 \left(c(z, t) - \widehat{c}(z, t; h_*)\right)^2 \, \mathrm{d}z\right)^{1/2}. \tag{19}$$

In practice the error (19) is effectively bounded below by the so-called *statistical error*, which is defined to be the expected value of $\|c - \widehat{c}\|_2$ in the event that the ensemble $\{Z_t^{(i)}, i = 1, ..., N\}$ were sampled from the exact distribution $c(z, t)$ itself. It is important to emphasise that it is not possible, using our method, to investigate schemes with errors below the statistical error. The statistical error can of course be reduced by using a larger ensemble $N$, but convergence is slow as the dependency

is $N^{-1/5}$, as discussed in Appendix D where details are given.



## 4.2 Results

The main results, showing the performance of the six schemes described in Tables 2-3 over a wide range of time-steps $\Delta t$, are shown in Figs. 4-6. Figs. 4-6 detail the results for the constant $\tau$ test case, the stable ABL test case and the neutral ABL test case respectively (see Table 1). In each figure, the $L_2$-error (19) is plotted as a function of non-dimensional time-step $\Delta t\, u_*/h$.

Logarithmic scales are used so that lines of constant slope corresponds to the observed order of the schemes. Blue lines with slopes 1, 2 and 3 are plotted for reference. The statistical error, which is the lowest possible error that can be measured for a given scheme, is plotted as a solid black line on each panel.

Also plotted on Figs. 4-6, as a dotted black line, is the $L_2$-norm difference $\|c - C_0\|_2$ between the concentration field $c(z,t)$ obtained from the solution of the FPE (3) and $C_0(z,t)$ obtained from the diffusion equation (15). The dotted black line marks

an important boundary on each panel. If the time-step $\Delta t$ is such that the error of a given scheme lies above this line, then it is preferable to solve the RDM (16) in place of (2), because (at fixed $\Delta t$) the numerical error for the former is more easily controlled.

Fig. 4 shows results for the constant $\tau$ test-case at time $t = 1$ $(h/u_*)$ (see Fig. 3 and Table 1 for details). The lowest order schemes, LEGGRAUP (blue circles) and E-M (black squares) are seen to realise their formal weak order $\Delta t$. EXPLICIT 2.0

(red hexagons) and HON-SRKII (green solid triangles) have weak order $\Delta t^2$, whereas EXPLICIT 3.0 (blue triangles) has weak order $\Delta t^3$ as expected. The best performing scheme for this particular case is the new scheme LONGSTEP (purple diamonds) derived in Appendix C. The rationale for LONGSTEP is that there is a conceptual error in the derivation of LEGGRAUP, which results in its performance at large $\Delta t$ being no better than E-M. When this error is corrected in LONGSTEP, the performance is better than even EXPLICIT 3.0. LONGSTEP in effect uses exact solutions of the LPDM equations for constant $\tau$ and

linear $\sigma_w$, meaning that if the same calculations had been performed in an infinite domain, the numerical error would be zero. In the constant $\tau$ test case, errors in LONGSTEP arise only from the reflection boundary conditions at $z = 0, 1$. However, LONGSTEP does not fare well in the remaining two test-cases to be described next.

Fig. 5 shows results for the stable ABL test case at intermediate time $t = 1\, h/u_*$ (upper panel) and at late time $t = 4\, h/u_*$ (lower panel), when the concentration is almost well-mixed across the ABL (see Fig. 3). The results are similar to those of

the constant $\tau$ case, except LONGSTEP (purple diamonds) now performs as poorly as E-M. Both E-M and LONGSTEP outperform LEGGRAUP. Note that physically acceptable solutions using the EXPLICIT 3.0 scheme at time-steps longer than $\Delta t = 0.02$, could not be resolved due to the reflection boundary conditions, as explained below.

Fig. 6 shows the results for the neutral ABL case at intermediate time $t = 3\, h/u_*$ (upper panel) and at late time $t = 12\, h/u_*$ (lower panel). In this case the performance of LONGSTEP and LEGGRAUP are comparable, but with the E-M scheme per-

forming better than both, except at very long time-steps where LEGGRAUP having slightly better accuracy at long time-steps. As for the previous test cases EXPLICIT 3.0 (blue triangles) scheme gives the lowest errors (weak order $\Delta t^3$), and EXPLICIT 2.0 (red hexagons) along with HON-SRKII (green solid triangles) perform consistently well with weak order $\Delta t^2$.

To give an impression of where the particle concentration errors are accumulating, Fig. 7 shows snapshots of particle density $\hat{c}(z,t)$ for the stable ABL case, at $t = h/u_*$. Results are shown for each scheme when a long time-step $\Delta t = 0.05\, h/u_*$ is used



(left panel) and a moderate time-step $\Delta t = 0.007\, h/u_*$ (right panel). The errors in the long time-step case are large and are largely due to issues with the reflection of trajectories at the surface ($z = 0$). Numerical accuracy requires that $\Delta t \ll \tau$, which is evidently violated close to the boundary where $\tau(z)$ is small (see Fig. 1). Errors due to reflection are particularly acute for the higher order schemes (such as EXPLICIT 2.0 and HON-SRKII) that require the treatment of an intermediate step(s). See

the discussion in Appendix B for how this step is implemented. The stable boundary layer case, where $\tau$ decays most rapidly near the $z = 0$ boundary, is the case which appears to be the most sensitive to the treatment of reflection there.

## 5    Conclusions

The main contribution of this paper is to introduce a protocol for the quantitative assessment of SDE numerical schemes, applied to the problem of dispersion in an idealised atmospheric boundary layer, as modelled by LPDMs. Accurate solutions

of the Fokker-Planck equation (FPE, 3) are used to benchmark the distribution obtained from an ensemble of LPDM solutions obtained using a particular scheme with a fixed time-step $\Delta t$. By using the FPE solution, our protocol avoids the possibility of the LPDMs exhibiting spurious convergence to an incorrect distribution as $\Delta t \to 0$ (e.g. by a poor treatment of reflection boundary conditions), and the FPE provides independent verification of the correctness of a specific implementation.

The convergence results obtained in our model test problems are valuable because, due to the importance of reflection of

particles from the surface and top of the boundary layer, it is not possible to rely on the formal convergence rates of different SDE schemes (as given by e.g. Kloeden and Platen, 1992). All of the schemes tested attain their formal convergence rates at early times in the model test problem, i.e. before reflection becomes important, and thereafter are limited to an extent by the details of how reflection is implemented (see Appendix B for discussion).

Our results allow the following recommendations to be made, for consideration by operational modellers:

– For our test problems, the best results with respect to accuracy as a function of $\Delta t$ were obtained with the weak order $\Delta t^3$ scheme EXPLICIT 3.0. However, this scheme is time-consuming to implement and more expensive per step compared to the weak order $\Delta t^2$ schemes investigated, so the gains associated with it are marginal. A good compromise between ease-of-implementation, flexibility and accuracy is the 'small-noise' scheme of (Honeycutt, 1992, here HON-SRKII). Formally, the weak order of HON-SRKII is just $\Delta t$, i.e. the same as Euler-Maruyama. However, the scheme designed so

that at fixed $\Delta t$, in the limit of small-noise the weak error scales with $\Delta t^2$ (e.g. Chap. 3 of Milstein and Tretyakov, 2004). Although the boundary layer dispersion problems examined here are not formally 'small-noise' problems, our results show clearly that they behave as such in a practical implementation. As a consequence HON-SRKII scheme performs at least as well as the formally weak order $\Delta t^2$ scheme EXPLICIT 2.0 (which in fact has a very similar implementation for the specific LPDM problem we have examined here).

– The 'long-step' scheme due to Legg and Raupach (1982, here LEGGRAUP), which is used operationally for global integrations of trajectories in FLEXPART (for example), should be avoided. LEGGRAUP does not significantly outperform Euler-Maruyama at any time-step for any of the three profiles we have studied. The reason for this is a conceptual





error in its derivation, which we have corrected here in the development of a new scheme LONGSTEP, see Appendix C. LONGSTEP performs very well in the case of $\tau(z) =$ constant, but no better than LEGGRAUP for other $\tau(z)$ profiles, hence we do recommend it for operational use either.

- – Global calculations often require the use of long time-steps for reasons of computational efficiency. For such calculations,
we recommend switching to the random displacement model (16), rather than solving the LPDM equations (2). The reason for this recommendation is apparent in Figs. 4-6. where the numerical error for all of the schemes investigated is seen to exceed the difference between RDM and LPDM solutions when the time-step $\Delta t \gtrsim 0.02$. Given that the unit of time in our non-dimensionalisation is $T = h/u_*$, where $h = 100-1000$m is boundary layer height and $u_* = 0.1-1\mathrm{ms}^{-1}$ is surface friction velocity, for a typical $T \approx 1000$s errors will be minimized by using the RDM whenever a time-step
$\Delta t \gtrsim 20$s is required.

Naturally, the recommendations above are based only on the limited set of schemes which we have studied. It is to be hoped that the protocol and test cases introduced here will be helpful to other researchers developing and testing novel methods for LPDMs. A key challenge in such development will be the careful treatment of reflection boundary conditions, including their generalisation to more complex physical situations (e.g. Wilson and Flesch, 1993; Thomson et al., 1997; Wilson and Yee,
15    2007).

## 6 Code availability

The MATLAB source code of the FPE solver can be found online via GitHub and by searching for the repository "MRE FPE solver" (https://github.com/nhramli/MRE-FPEsolver.git).

## Appendix A: Useful properties of (probabilists') Hermite polynomials and functions

In this appendix we detail some useful properties of the probabilists' Hermite polynomials $\mathrm{He}_k(\omega)$, defined by (8), and the associated Hermite functions $\mathrm{He}_k(\omega)\mathrm{e}^{-\omega^2/2}/\sqrt{2\pi}$. We concentrate on those identities necessary to derive (11), all can be obtained easily from results found in (see Chapter 22 of Abramowitz and Stegun, 1965).

First, the Hermite polynomials are solutions of Hermite's equation

$$\left(\frac{\partial^2}{\partial\omega^2} - \omega\frac{\partial}{\partial\omega}\right)\mathrm{He}_k(\omega) = -k\mathrm{He}_k(\omega), \tag{A1}$$

from which it follows that the Hermite functions satisfy

$$\left(\frac{\partial^2}{\partial\omega^2} + \omega\frac{\partial}{\partial\omega} + 1\right)\left(\frac{\mathrm{He}_k(\omega)\mathrm{e}^{-\omega^2/2}}{\sqrt{2\pi}}\right) = -k\frac{\mathrm{He}_k(\omega)\mathrm{e}^{-\omega^2/2}}{\sqrt{2\pi}} \tag{A2}$$





Second, because Hermite's equation can be written as an eigenvalue problem with a self-adjoint linear operator, the Hermite polynomials can be shown to satisfy an orthogonality relation, specifically

$$\int\limits_{-\infty}^{\infty} \mathrm{He}_j(\omega)\mathrm{He}_k(\omega)\frac{\mathrm{e}^{-\omega^2/2}}{\sqrt{2\pi}}\mathrm{d}\omega = k!\,\delta_{jk}, \tag{A3}$$

where $\delta_{jk}$ is the Kronecker delta. Thirdly and fourthly, the following differentiation and recursion relations can be obtained

$$\frac{\mathrm{d}}{\mathrm{d}\omega}\mathrm{He}_k(\omega) = k\mathrm{He}_{k-1}(\omega) \tag{A4}$$

$$\omega\mathrm{He}_k(\omega) = \mathrm{He}_{k+1}(\omega) + k\mathrm{He}_{k-1}(\omega). \tag{A5}$$

The results (A2-A5) are used in the derivation of (11).

### Appendix B: Boundary condition implementation

#### B1 FPE numerical boundary conditions

The FPE (3) is solved numerically by integrating (11) using the central finite difference method

$$\frac{\partial C_k}{\partial t}(z_i) = -\frac{k}{\tau(z_i)}C_k(z_i) - (k+1)\frac{\sigma_w(z_{i+1})C_{k+1}(z_{i+1}) - \sigma_w(z_{i-1})C_{k+1}(z_{i-1})}{2\Delta z}$$
$$- \sigma_w(z_i)\frac{C_{k-1}(z_{i+1}) - C_{k-1}(z_{i-1})}{2\Delta z} \tag{B1}$$

where $\Delta z = 1/N_z$ and $z_i = (i-1/2)\Delta z$, for $i = 1,\ldots,N_z$. Careful treatment is necessary at the boundaries. For $k$ odd, the
physical boundary conditions $C_k(0,t) = C_k(1,t) = 0$ imply the following substitutions for the values at the virtual points at
$z = z_0$ and $z = z_{N_z+1}$, $C_k(z_0) = -C_k(z_1)$ and $C_k(z_{N_z+1}) = -C_k(z_{N_z})$. For $k$ even, the equation itself with $k$ odd requires
$C_k(z_0) = C_k(z_1)$ and $C_k(z_{N_z+1}) = C_k(z_{N_z})$. These substitutions allow the right-hand side of (B1) to be expressed as a $N_zK \times$
$N_zK$ matrix equation and completes the discretisation.

#### B2 LPDM numerical boundary conditions

In the numerical implementation of LPDM (2), the reflection condition $\Omega_t \to -\Omega_t$ is applied at the bottom and top of the
ABL, where $Z_t = 0$ and $Z_t = 1$ respectively. This means that perfect reflection at the boundaries is also assumed for the $Z_t$
computation.

- At the end of every time step of the numerical scheme $t_j = j\,\Delta t$ $(j = 1, 2, 3, \ldots)$, any 'illegal' particle $(\Omega_{t_j}^*, Z_{t_j}^*)$ that
  crosses the boundaries, i.e. below $Z = 0$ or above $Z = 1$, will be reflected back into the domain and its velocity direction
is reversed, i.e. $\Omega_{t_j} = -\Omega_{t_j}^*$.

- Higher-order schemes involve intermediate time-steps. Our treatment of intermediate time-steps is as follows. First, the
  $z$-domain is extended to $z \in (-\infty, \infty)$, by repeated reflection of the $\sigma_w(z)$ and $\tau(z)$ profiles in the boundaries. In this





extended domain, all intermediate time-steps are completed according to the algorithm in question. Then, at the end of the completed time-step reflection, as detailed above, takes the particle back into the $z \in [0, 1]$ domain where necessary. The domain extension device thus uniquely determines an unambiguous treatment of reflection of particles near the boundaries in the higher weak order schemes EXPLICIT 2.0, HON-SRKII, EXPLICIT 3.0 detailed in Tables 2-3.

## 5  Appendix C: Derivation of a new long time-stepping scheme (LONGSTEP)

Here we derive a new long time-step scheme LONGSTEP. The scheme is designed to give acceptable results when integrating (2) using time-steps $\Delta t \gtrsim \mathrm{Min}\,\tau(z)$, for use in operational models. The starting point for the scheme is the velocity update equation for the LEGGRAUP scheme (see Table 2)

$$\Omega_{n+1} = R_n \Omega_n + \sigma'_n \tau_n (1 - R_n) + \left(1 - R_n^2\right)^{1/2} \Delta_n, \tag{C1}$$

where $\tau_n = \tau(Z_n), \sigma'_n = (\partial \sigma_w / \partial z)(Z_n), R_n = \exp\left(-\Delta t / \tau_n\right)$ and $\Delta_n \sim \mathcal{N}(0, 1)$ is a random variable drawn from a Gaussian distribution with zero mean and unit variance. This scheme is obtained by first transforming (2) using Itô's lemma to obtain

$$\mathrm{d}\left(\Omega_t \mathrm{e}^{t/\tau}\right) = \mathrm{e}^{t/\tau} \frac{\partial \sigma_w}{\partial z} \, \mathrm{d}t + \mathrm{e}^{t/\tau} \left(\frac{2}{\tau}\right)^{1/2} \mathrm{d}B_t.$$

If both $\tau$ and $\partial \sigma_w / \partial z$ are taken to be constant (i.e. $\sigma_w(z) = \sigma_0 + \sigma'_0 z$), this equation can be integrated to give

$$\Omega_t = \Omega_0 \mathrm{e}^{-t/\tau} + \sigma'_0 \tau(z) \left(1 - \mathrm{e}^{-t/\tau}\right) + \left(\frac{2}{\tau}\right)^{1/2} \int_0^t \mathrm{e}^{(s-t)/\tau} \, \mathrm{d}B_s. \tag{C2}$$

Stochastic integrals of the form

$$\int_0^t f(s) \, \mathrm{d}B_s \sim \mathcal{N}\left(0, \int_0^t f(s)^2 \, \mathrm{d}s\right),$$

hence the final term in (C2) can be replaced by a Gaussian random variable to give

$$\Omega_t = \Omega_0 \mathrm{e}^{-t/\tau} + \sigma'_0 \tau \left(1 - \mathrm{e}^{-t/\tau}\right) + \alpha_1(t) \Delta_1, \tag{C3}$$

where $\Delta_1 \sim \mathcal{N}(0, 1)$ and $\alpha_1(t) = \left(1 - \mathrm{e}^{-2t/\tau}\right)^{1/2}$. Equation (C1) used by LEGGRAUP follows immediately from this solution.

The point where our analysis departs from that of Legg and Raupach (1982) is in the derivation of the position update. Under the approximation of linear $\sigma_w$ the position equation of (2) is

$$\mathrm{d}Z_t = \Omega_t \left(\sigma_0 + \sigma'_0 Z_t\right) \mathrm{d}t, \tag{C4}$$

which, applying Itô's lemma, can be written as

$$\mathrm{d}\left(\log\left(\sigma_0 + \sigma'_0 Z_t\right)\right) = \sigma'_0 \Omega_t \, \mathrm{d}t \tag{C5}$$





and integrated to obtain

$$\frac{1}{\sigma_0'} \left( \log \left( \sigma_0 + \sigma_0' Z_t \right) - \log \left( \sigma_0 + \sigma_0' Z_0 \right) \right) = \int\limits_0^t \Omega_s \, \mathrm{d}s. \tag{C6}$$

The update equation used in LEGGRAUP, i.e. from Table 2,

$$Z_{n+1} = Z_n + \sigma_n \Omega_n \, \mathrm{d}t \tag{C7}$$

would be correct (only in the limit $\sigma_0' \to 0$) in the event that $\Omega_s$ were a deterministic variable in the interval $0 \le s \le t$. However, $\Omega_s$ is a stochastic variable, hence it is a very crude approximation (error $O(t)$) to replace the integral on the right-hand side of (C6) by $\Omega_0 t$ (which leads to the update eqn. C7). Instead, the integral needs to be considered carefully, as follows.

To evaluate the *stochastic* integral on the right-hand side of (C6) integral we can insert the solution (C2) for $\Omega_s$ to obtain

$$\int\limits_0^t \Omega_s \, \mathrm{d}s = \Omega_0 \tau \left( 1 - \mathrm{e}^{-t/\tau} \right) + \sigma_0' \tau^2 \left( \frac{t}{\tau} - 1 + \mathrm{e}^{-t/\tau} \right) + \left( \frac{2}{\tau} \right)^{1/2} \int\limits_0^t \int\limits_0^s \mathrm{e}^{(q-s)/\tau} \, \mathrm{d}B_q \, \mathrm{d}s, \tag{C8}$$

The final term can be evaluated following a switch in the order of integration

$$\int\limits_0^t \int\limits_0^s \mathrm{e}^{(q-s)/\tau} \, \mathrm{d}B_q \, \mathrm{d}s = \int\limits_0^t \int\limits_q^t \mathrm{e}^{(q-s)/\tau} \, \mathrm{d}s \, \mathrm{d}B_q = \tau \int\limits_0^t \left( 1 - \mathrm{e}^{(q-t)/\tau} \right) \mathrm{d}B_q = \tau^{3/2} \alpha_2(t) \hat{\Delta}_2. \tag{C9}$$

where $\hat{\Delta}_2 \sim \mathcal{N}(0,1)$ and

$$\alpha_2(t) = \left( \frac{t}{\tau} - 2 \left( 1 - \mathrm{e}^{-t/\tau} \right) + \frac{1}{2} \left( 1 - \mathrm{e}^{-2t/\tau} \right) \right)^{1/2}.$$

The issue for implementation is that the Gaussian random variables $\Delta_1$ and $\hat{\Delta}_2$ are not independent. In fact, they have covariance given by

$$\mathbb{E}(\Delta_1 \hat{\Delta}_2) \equiv \beta(t) = \frac{\sqrt{2}}{\tau^2 \alpha_1(t) \alpha_2(t)} \int\limits_0^t \mathrm{e}^{(s-t)/\tau} \left( 1 - \mathrm{e}^{(s-t)/\tau} \right) \mathrm{d}s = \frac{\left( 1 - \mathrm{e}^{-t/\tau} \right)^2}{\sqrt{2} \, \alpha_1(t) \, \alpha_2(t)}.$$

Independent random variables can be introduced by writing

$$\hat{\Delta}_2 = \beta(t) \, \Delta_1 + \left( 1 - \beta(t)^2 \right)^{1/2} \Delta_2, \tag{C10}$$

where $\Delta_1$ and $\Delta_2$ are independent with $\Delta_1, \Delta_2 \sim \mathcal{N}(0,1)$.

The explicit solution of (C6) can therefore be written

$$Z_t = Z_0 \exp \left( \sigma_0' S_0 \right) + \frac{\sigma_0}{\sigma_0'} \left( \exp \left( \sigma_0' S_0 \right) - 1 \right), \quad \text{where} \tag{C11}$$

$$S_0 = \Omega_0 \tau \left( 1 - \mathrm{e}^{-t/\tau} \right) + \sigma_0' \tau^2 \left( \frac{t}{\tau} - 1 + \mathrm{e}^{-t/\tau} \right) + 2^{1/2} \alpha_2(t) \left( \beta(t) \, \Delta_1 + \left( 1 - \beta(t) \right)^{1/2} \Delta_2 \right)$$

The scheme LONGSTEP, given explicitly in Table 2, consists of the LEGGRAUP velocity update (C1), and a position update obtained from the solution (C11) by linearizing $\sigma_w$ about the current position $Z_n$. Similar to E-M, LONGSTEP converges with weak error $\sim \Delta t$, however it is designed to perform better at long time-steps as will be tested in section 5.





## Appendix D: Bandwidth selection and statistical error

Physical concentrations of the particles, $\widehat{c}(z,t)$ can be reconstructed from the resulting LPDM trajectory ensemble solution, $\{Z_t^{(i)}, i = 1, ..., N\}$ using the kernel density estimation (KDE) (e.g. Silverman, 1986; Wand and Jones, 1994). KDE is a statistical technique used to estimate an unknown pdf from a finite set of $N$ independent samples drawn from a random variable with that same pdf. KDE works by weighting each particle by a smoothed kernel function so that the probability represented by each particle becomes continuously spread out in space, see equation (18).

Once a sensible kernel function is chosen, and here we use the Gaussian kernel $K_G(x) = \mathrm{e}^{-x^2/2}/\sqrt{2\pi}$, the challenge in KDE is to choose the optimal kernel bandwidth $h = h_*$ that will result in the most accurate reconstruction of the pdf. Suppose that $c(z)$ is a pdf and $\widehat{c}(z, h)$ is a reconstruction of $c(z)$ based on $N$ samples drawn independently from a random variable with pdf $c(z)$, using (18) with bandwidth $h$. The optimal $h_*$ is then typically chosen to minimise the expected value of $\|c - \widehat{c}\|_2^2$ i.e. the square of the $L_2$-error norm (often called the Mean-Integrated-Square-Error or MISE in the statistics literature). The key result (see e.g. Silverman, 1986) is that

$$h_* = N^{-1/5} I^{-2/5} \alpha^{-2/5} \beta^{1/5}$$

where $I = \int_0^1 c_{zz}(z,t)^2 \, \mathrm{d}z$, $\beta = \int_{-\infty}^{\infty} K_G(z)^2 \, \mathrm{d}z = 1/\sqrt{4\pi}$ and $\alpha = \int_{-\infty}^{\infty} z^2 K_G(z) \, \mathrm{d}z = 1$. The choice $h = h_*$ results in the expected minimum error being

$$\mathrm{Min}\|c - \widehat{c}\|_2^2 = \frac{5}{4} \beta^{4/5} \alpha^{2/5} I^{1/5} N^{-4/5}, \tag{D1}$$

which is the formula for statistical error which allows the solid line to be calculated and plotted in Figs. 4-6.

*Acknowledgements.* JGE acknowledges support from UK Natural Environment Research Council grant NE/G016003/1. HMR acknowledges support from UBD Chancellor's Scholarship and UCL Studentship.



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

| | $\sigma_w(z)$ | $\tau(z)$ | Modified $\bar{\sigma}_w(z)$ | Modified $\bar{\tau}(z)$ |
|---|---|---|---|---|
| Constant $\tau$ | $0.5\,(1+z)$ | constant | – | – |
| Stable | $1.3\,(1-z)$ | $\dfrac{0.1z^{4/5}}{\sigma_w}$ | $\sigma_w(Z_m(z))$ | $\tau(Z_m(z))$ |
| Neutral | $1.3\exp(-2z/\epsilon)$ | $\dfrac{z}{2\sigma_w(1+15z/\epsilon)}$ | $\sigma_w(Z_m(z))$ | $\tau(Z_m(z))$ |

**Table 1.** The non-dimensional profiles of $\sigma_w(z)$ and $\tau(z)$ suitable for (i) a constant $\tau$ profile, (ii) a stable ABL, and (iii) a neutral ABL (e.g. Hanna, 1982). The non-dimensional parameter $\epsilon = u_*/fh$ is a boundary layer Rossby number (the value $\epsilon = 0.8$ is taken in the test case). For the purposes of numerical stability (see text), in practice the modified profiles $\bar{\sigma}_w(z)$ and $\bar{\tau}(z)$ are used, where $Z_m(z) = z_b + z(1-2z_b)$ is chosen to avoid singular behaviour at the boundaries ($z_b = 0.05$).



| Scheme | Algorithm | Reference and Notes |
|---|---|---|
| E-M | $\Omega_{n+1} = \Omega_n + F_n\,\Delta t + (2/\tau_n)^{1/2}\,\Delta B_n$ <br> $Z_{n+1} = Z_n + \Omega_n\sigma_n\,\Delta t$ | Maruyama (1955) |
| EXPLICIT 2.0 | $\Omega_{n+1} = \Omega_n + \frac{1}{2}\left(F_n + F_\mu\right)\Delta t + \frac{1}{2}\left((2/\tau_n)^{1/2} + (2/\tau_\mu)^{1/2}\right)\Delta B_n$ <br> $Z_{n+1} = Z_n + \frac{1}{2}\left(\Omega_n\sigma_n + \Omega_\mu\sigma_\mu\right)\Delta t$ <br> $\quad\Omega_\mu = \Omega_n + F_n\,\Delta t + (2/\tau_n)^{1/2}\,\Delta B_n,$ <br> $\quad\; Z_\mu = Z_n + \Omega_n\sigma_n\,\Delta t$ | Sec. 15.1 of Kloeden and Platen (1992) |
| HON-SRKII | $\Omega_{n+1} = \Omega_n + \frac{1}{2}\left(F_n + F_\mu\right)\Delta t + (2/\tau_n)^{1/2}\,\Delta B_n$ <br> $Z_{n+1} = Z_n + \frac{1}{2}\left(\Omega_n\sigma_n + \Omega_\mu\sigma_\mu\right)\Delta t$ <br> $\quad\Omega_\mu = \Omega_n + F_n\,\Delta t + (2/\tau_n)^{1/2}\,\Delta B_n,$ <br> $\quad\; Z_\mu = Z_n + \Omega_n\sigma_n\,\Delta t$ | Honeycutt (1992) |
| LEGGRAUP | $\Omega_{n+1} = R_n\,\Omega_n + \sigma'_n\,\tau\,(1 - R_n) + \left(1 - R_n^2\right)^{1/2}\Delta_n$ <br> $Z_{n+1} = Z_n + \sigma_n\,\Omega_n\,\Delta_n$ <br> $\quad R_n = \mathrm{e}^{-\Delta t/\tau_n}$ | Legg and Raupach (1982) |
| LONGSTEP | $\Omega_{n+1} = R_n\,\Omega_n + \sigma'_n\,\tau\,(1 - R_n) + \left(1 - R_n^2\right)^{1/2}\Delta_n$ <br> $Z_{n+1} = Z_n + \frac{\sigma_n}{\sigma'_n}\left(\exp\left(\sigma'_n\,S_n\right) - 1\right)$ <br> $\quad R_n = \mathrm{e}^{-\Delta t/\tau_n}$ <br> $\quad S_n = \Omega_n\tau_n\left(1 - \mathrm{e}^{-\Delta t/\tau_n}\right) + \sigma'_n\tau_n^2\left(\frac{\Delta t}{\tau_n} - 1 + \mathrm{e}^{-\Delta t/\tau_n}\right)$ <br> $\qquad\qquad + 2^{1/2}\alpha_{2n}(t)\left(\beta_n\,\Delta_{1n} + (1 - \beta_n)^{1/2}\,\Delta_{2n}\right)$ <br> $\quad \beta_n = \frac{(1 - R_n)^2}{2^{1/2}\,\alpha_{1n}\alpha_{2n}}, \qquad \alpha_{1n} = (1 - R_n)^{1/2}$ <br> $\quad \alpha_{2n} = \left(\frac{\Delta t}{\tau_n} - 2(1 - R_n) + \frac{1}{2}\left(1 - R_n^2\right)\right)^{1/2}$ | See Appendix C |

**Table 2.** The LPDM numerical schemes investigated in section 4. Here $\Delta t$ is the time-step, $\Delta B_n \sim \mathcal{N}(0, \Delta t)$, $\Delta_n \sim \mathcal{N}(0,1)$ and $\sigma_i = \sigma_w(Z_i)$, $\tau_n = \tau(Z_n)$. The drift function is denoted by $F_i = -\Omega_i/\tau(Z_i) + \sigma'_w(Z_i)$ where $i = n, \mu$.



| Scheme | Algorithm | Reference and Notes |
|---|---|---|
| EXPLICIT 3.0 | $\Omega_{n+1} = \Omega_n + F_n\,\Delta t + (2/\tau_n)^{1/2}\Delta B_n$ $\quad + \frac{1}{2}\left(F_\zeta^+ + F_\zeta^- - \frac{3}{2}F_n - \frac{1}{4}\left(\tilde{F}_\zeta^+ + \tilde{F}_\zeta^-\right)\right)\Delta t$ $\quad + \left(\frac{1}{\sqrt{2}}\left(F_\zeta^+ - F_\zeta^-\right) - \frac{1}{4}\left(\tilde{F}_\zeta^+ - \tilde{F}_\zeta^-\right)\right)\varsigma\,\Delta C_n\,(2/\Delta t)^{1/2}$ $\quad + \frac{1}{6}\left(F_n + F_u - F_\zeta^+ - F_\rho^+\right)\left((\varsigma+\rho)\,\Delta B_n\,(\Delta t)^{1/2} + \Delta t + \varsigma\,\rho\left((\Delta B_n)^2 - \Delta t\right)\right)$ $Z_{n+1} = Z_n + \Omega_n\sigma_n\,\Delta t$ $\quad + \frac{1}{2}\left(\sigma_\zeta\left(\Omega_\zeta^+ + \Omega_\zeta^-\right) - \frac{3}{2}\Omega_n\sigma_n - \frac{1}{4}\tilde{\sigma}_\zeta\left(\tilde{\Omega}_\zeta^+ + \tilde{\Omega}_\zeta^-\right)\right)\Delta t$ $\quad + \left(\frac{\sigma_\zeta}{\sqrt{2}}\left(\Omega_\zeta^+ + \Omega_\zeta^-\right) - \frac{\tilde{\sigma}_\zeta}{4}\left(\tilde{\Omega}_\zeta^+ - \tilde{\Omega}_\zeta^-\right)\right)\varsigma\,\Delta C_n\,(2/\Delta t)^{1/2}$ $\quad + \frac{1}{6}\left(\Omega_n\sigma_n + \Omega_u\sigma_u - \sigma_\zeta\left(\Omega_\zeta^+ + \Omega_\rho^-\right)\right)\left((\varsigma+\rho)\,\Delta B_n\,(\Delta t)^{1/2} + \Delta t + \varsigma\,\rho\left((\Delta B_n)^2 - \Delta t\right)\right)$ $\;$ $\Omega_\phi^\pm = \Omega_n + F_n\,\Delta t \pm (2/\tau_n)^{1/2}(\Delta t)^{1/2}\phi$ $Z_\phi = Z_n + \Omega_n\sigma_n\,\Delta t$ $\tilde{\Omega}_\phi^\pm = \Omega_n + 2\,F_n\,\Delta t \pm (2/\tau_n)^{1/2}\,(2\,\Delta t)^{1/2}\,\phi$ $\tilde{Z}_\phi = Z_n + 2\,\Omega_n\sigma_n\,\Delta t$ $\Omega_u = \Omega_n + \left(F_n + F_\zeta^+\right)\Delta t + (2/\tau_n)^{1/2}\,(\varsigma+\rho)\,(\Delta t)^{1/2}$ $Z_u = Z_n + \left(\Omega_n\sigma_n + \Omega_\zeta^+\sigma_\zeta\right)\Delta t$ $\quad$ where $\phi = \zeta, \rho$ and $P(\zeta = \pm 1) = P(\rho = \pm 1) = \frac{1}{2}$ | Sec. 15.2 of Kloeden and Platen (1992) |

**Table 3.** EXPLICIT 3.0 scheme tested in section 4, with $\tau_n = \tau(Z_n)$, and $\sigma_i = \sigma_w(Z_i)$, $\tilde{\sigma}_\phi = \sigma_w(\tilde{Z}_\phi)$, where $i = n, u, \phi$. The drift function is denoted by $F_i = -\Omega_i/\tau(Z_i) + \sigma'_w(Z_i)$ or $\tilde{F}_\phi = -\tilde{\Omega}_i/\tau(\tilde{Z}_\phi) + \sigma'_w(\tilde{Z}_\phi)$. Here $\Delta t$ is the time step and we use two correlated Gaussian random variables $\Delta B_n \sim \mathcal{N}(0, \Delta t)$ and $\Delta C_n \sim \mathcal{N}(0, (\Delta t)^3/3)$, with $E(\Delta B_n \Delta C_n) = (\Delta t)^2/2$.





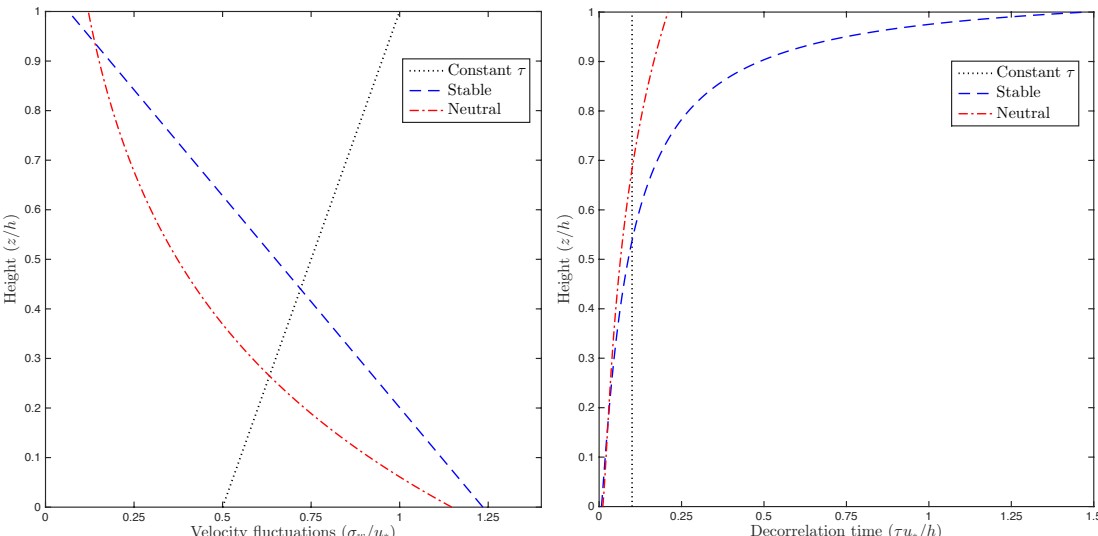

**Figure 1.** Vertical profiles of vertical velocity fluctuations $\bar{\sigma}_w(z)$ (left panel) and vertical velocity Lagrangian decorrelation time $\bar{\tau}(z)$ (right panel) used in the test-case problems (see Table 1). The dimensions for $\bar{\sigma}_w$ (right panel) and $\bar{\tau}$ are frictional velocity $u_*$ and $h/u_*$ respectively, where $h$ is the ABL height.



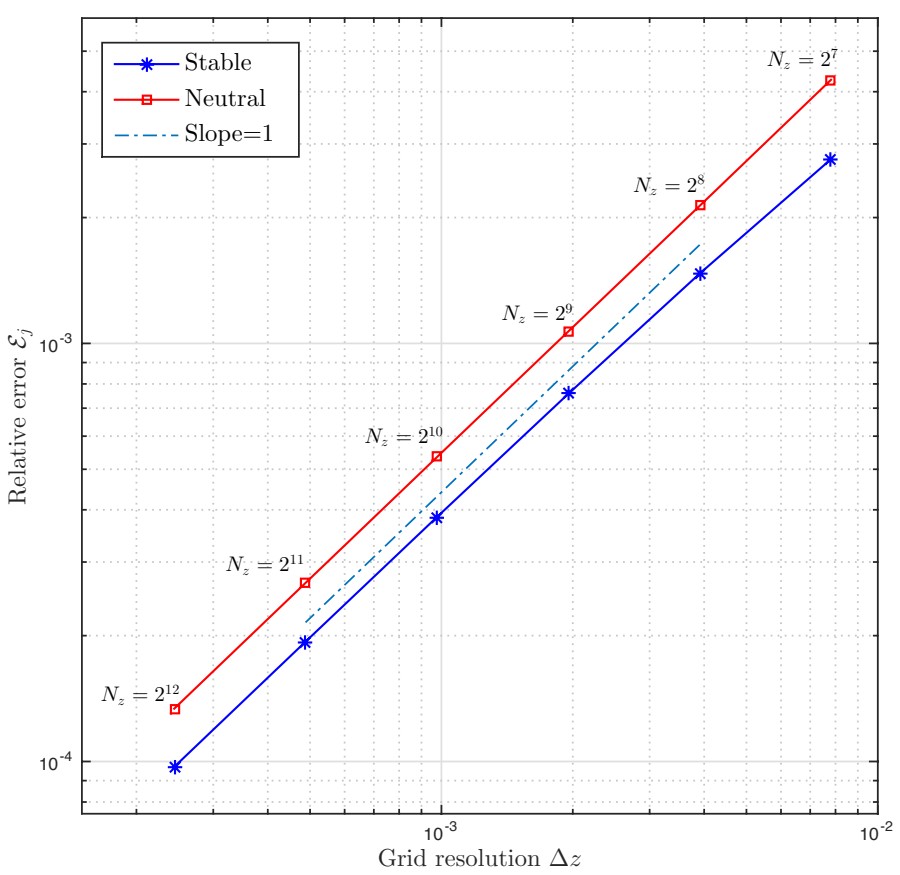

**Figure 2.** Relative error $\mathcal{E}_j$ (see eqn. 17) of the FPE solutions as a function of grid resolution $\Delta z = 2^{-j}$ for $j = 7, 8, ..., 12$ for the two test-case problems. Stars: Stable ABL ($\mathcal{E}_j(t=1)$). Squares: Neutral ABL ($\mathcal{E}_j(t=3)$).





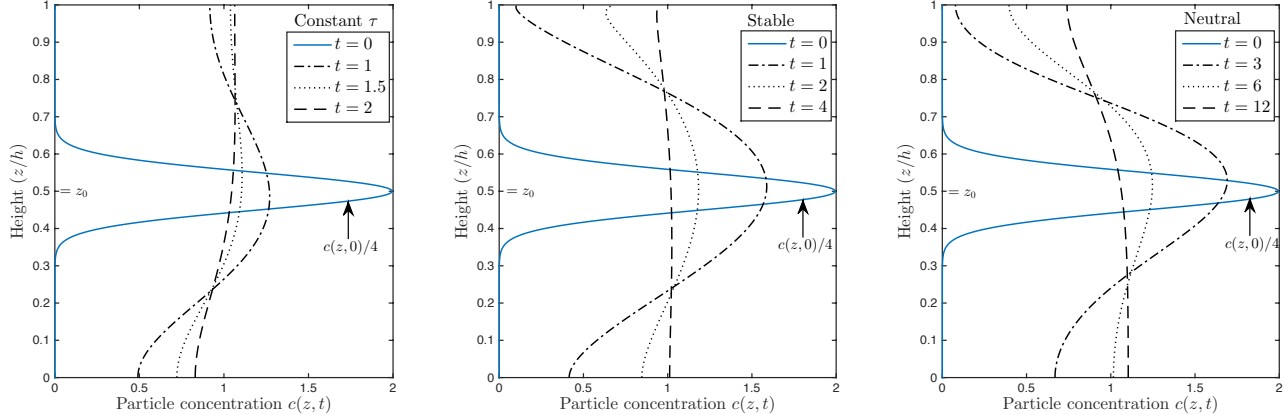

**Figure 3.** Snapshots of particle concentration $c(z,t)$ from the numerical FPE solutions for the three test-case problems. Left: Constant $\tau$ ($t = 0, 1, 1.5, 2\ h/u_*$). Center: Stable ABL ($t = 0, 1, 2, 4\ h/u_*$). Right: Neutral ABL ($t = 0, 3, 6, 12\ h/u_*$). For clarity $c(z,0)/4$ is plotted (instead of $c(z,0)$) for the initial condition at $t = 0$ in both panels.

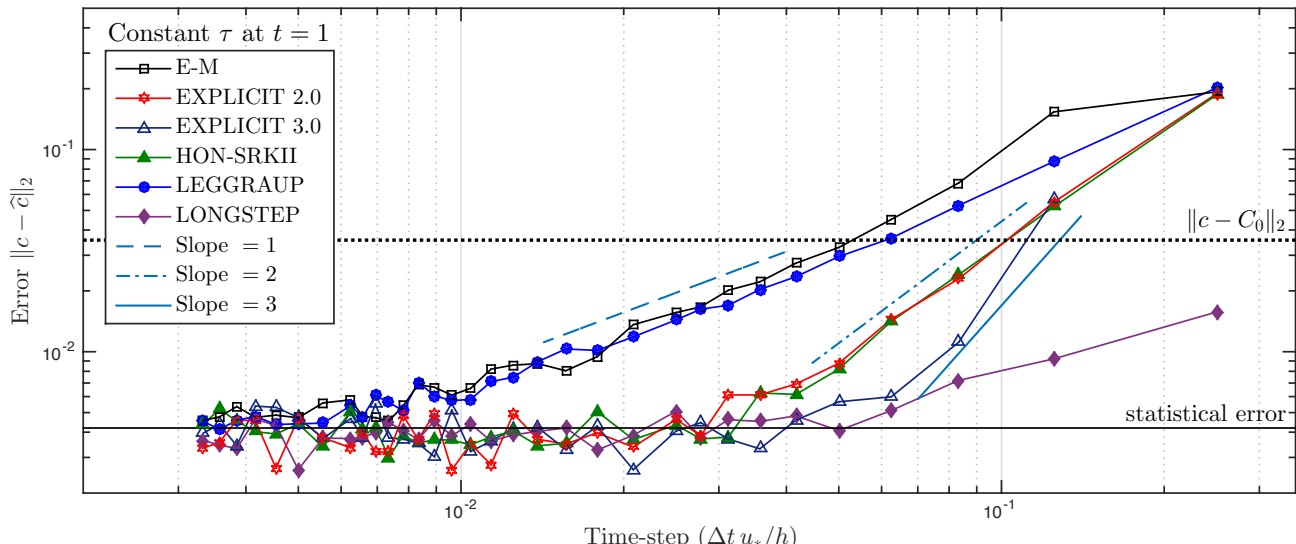

**Figure 4.** $L_2$-error (19) as a function of non-dimensional time-step $\Delta t\, u_*/h$ for the constant $\tau = 0.1$ test-case with $N = 10^6$ ensemble integrated at time $t = 1(h/u_*)$. The LONGSTEP scheme (purple diamonds) gives the best results in this case. Blue lines of slopes 1, 2 and 3 are plotted for reference.



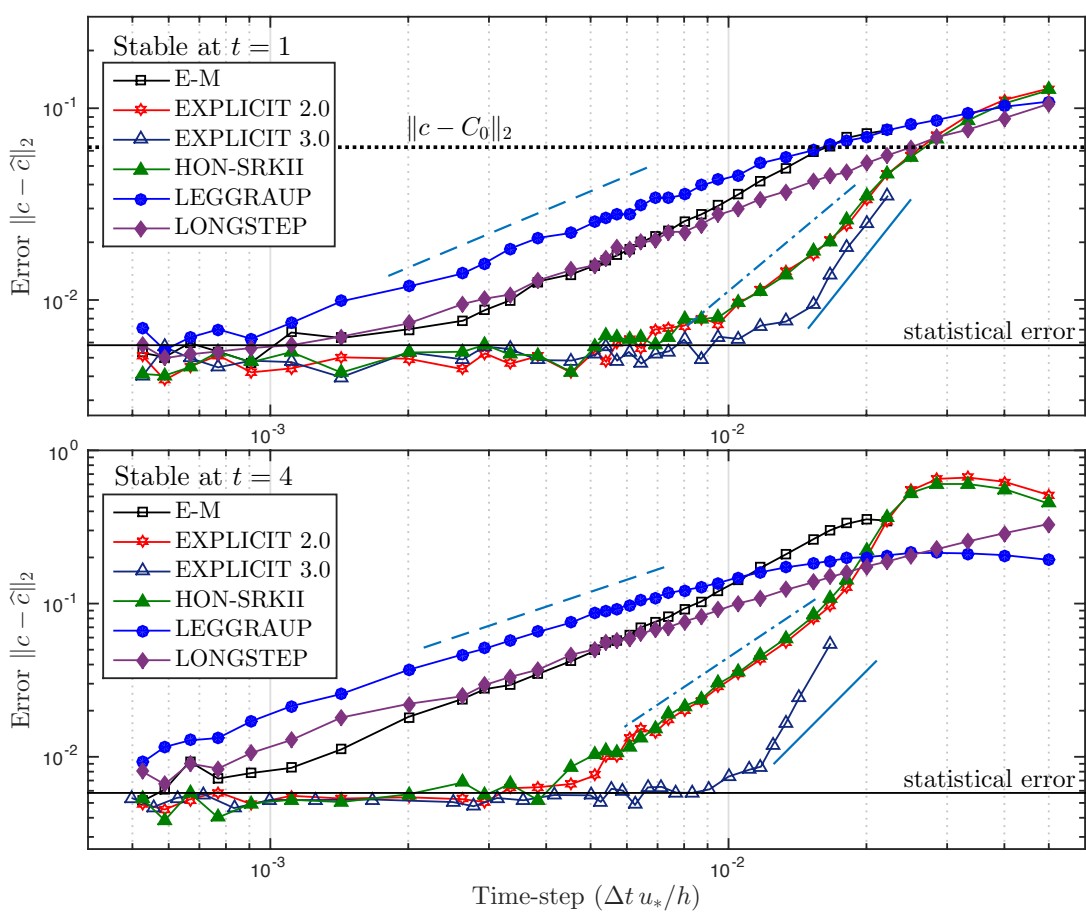

**Figure 5.** $L_2$-error (19) as a function of non-dimensional time-step $\Delta t\, u_*/h$ for the stable ABL test case integrated at intermediate time $t = 1\,h/u_*$ (upper panel) and at late time $t = 4\,h/u_*$ (lower panel). From left to right, blue lines of slopes $1, 2$ and $3$ are plotted for reference.



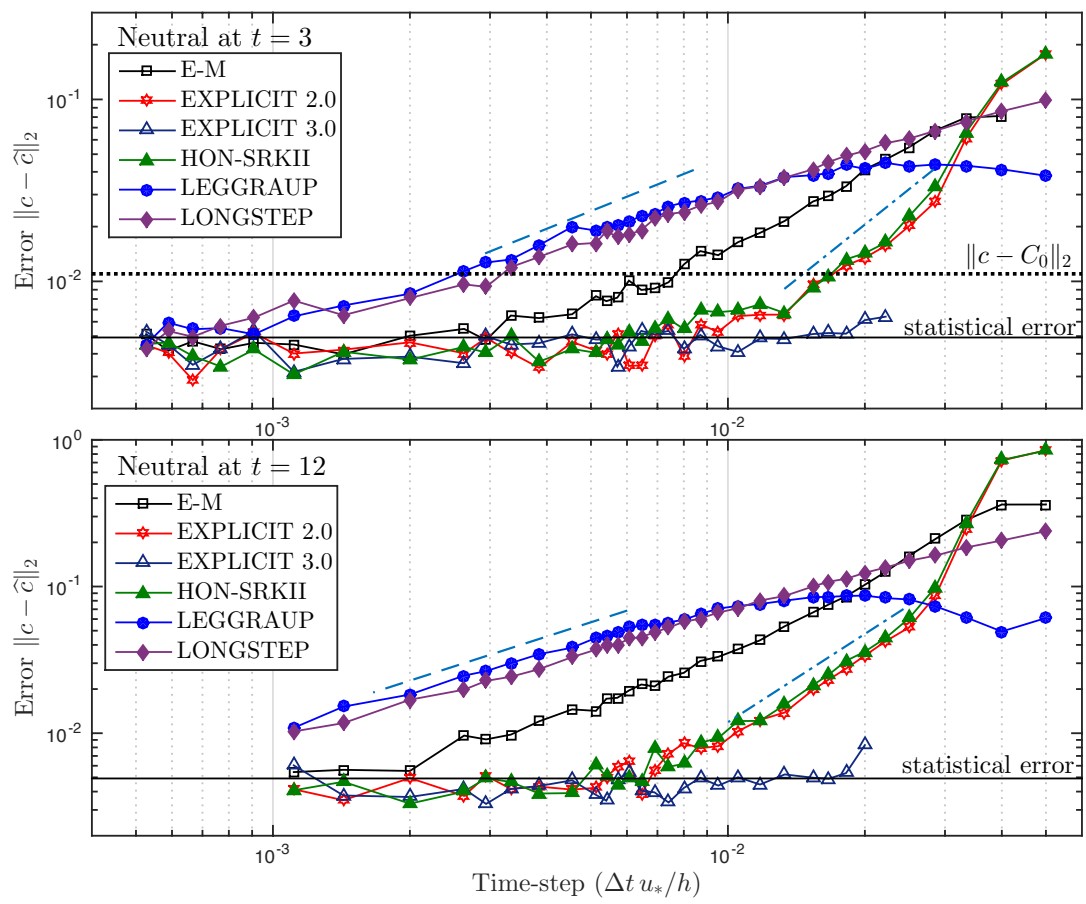

**Figure 6.** $L_2$-error (19) as a function of non-dimensional time-step $\Delta t\, u_*/h$ for the neutral ABL test case integrated at intermediate time $t = 3\, h/u_*$ (upper panel) and at late time $t = 12\, h/u_*$ (lower panel). From left to right, blue lines of slopes $1, 2$ and $3$ are plotted for reference.





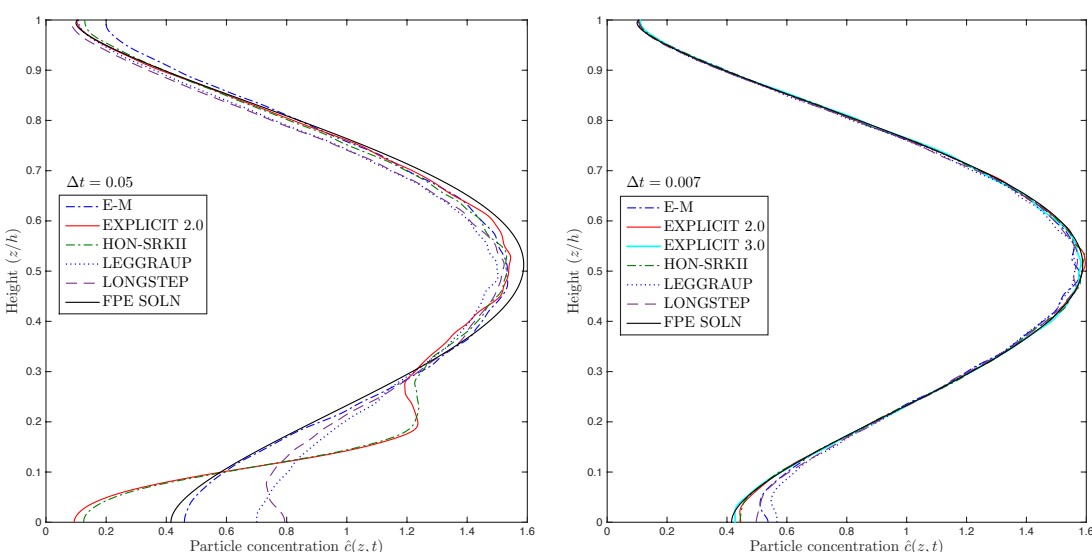

**Figure 7.** Snapshots of reconstructed particle density $\widehat{c}(z,t)$ for the stable ABL case at time $t = h/u_*$, shown at each scheme. Left: when long time-step $\Delta t = 0.05\,h/u_*$ is used and errors due to boundary conditions dominate. Right: when moderate time-step $\Delta t = 0.007\,h/u_*$ is used.