# Peer review of "Quantitative evaluation of numerical integration schemes for Lagrangian particle dispersion models"

_Geoscientific Model Development, 2016_

## Referee Comment (RC1) · Anonymous Referee #1 · 22 Apr 2016

**Comments on 'Quantitative evaluation of numerical integration schemes for Lagrangian particle dispersion models'**

This is a well written and informative paper concerning the numerical integration of a class of stochastic differential equations commonly used in atmospheric dispersion models using different numerical methods. These methods are compared both with each other and the results of the corresponding Fokker-Planck equation and the numerical solution of an appropriate diffusion equation (or random walk model). The authors' results provide a useful benchmark for selecting an appropriate numerical method and as such are likely to have widespread application. I have a few minor comments.

**Minor comments**

p.4 End of first line of §2.2 (l.6): delete extra 'the'.

p.4 Beginning of §2.2: it would be useful to define $\omega$.

p.5 l.16: I think it would be useful to state explicitly what the initial conditions for $C_k$ are.

p.6 Regarding equation (6) and the preceding text: the random walk model or diffusion equation is only a well-justified approximation of a Lagrangian particle dispersion model (equation (1)) for small $\tau$.

p.9 Line 23 and again in the caption to figure 5: is it necessary to include '1' in $t = 1h/u_*$?

p.11 Appendix A: you may wish to consider quoting the result for $\int_{-\infty}^{\infty} \mathrm{e}^{-\omega^2/2} \mathrm{He}_k(\omega) \, \mathrm{d}\omega$ which I found useful.

Fig. 1 Fig. 1 is not referred to until p. 10 (l. 3). Did the authors mean to refer to the figure earlier in the study?

---

## Referee Comment (RC2) · Anonymous Referee #2 · 22 May 2016

This is an interesting manuscript. It investigates the numerical solution of the stochastic differential equations (SDE) used for the Lagrangian vertical velocity in many operational Lagrangian stochastic (LS) models for turbulent dispersion. This is an aspect that has been often overlooked and a careful investigation is welcome and useful. Several schemes are compared and the authors even propose an original improvement/correction of a previously proposed method (LR 1982) used for long time-steps. However, their results suggest that, for long time-steps, the random displacement model may be a better model compared to the use of the SDE for the particle velocity. The manuscript certainly deserves to be published and I have only some minor comments that the authors should consider.

1) The calculation time (in seconds) required for any computational scheme should be given and discussed. This is a fundamental aspect and needs to be well clarified (for example in a table). 2) The authors consider only fixed time-step, while often LS models use a variable time-step (often linked to the Lagrangian integral time scale with some additional constraints). The authors should discuss this aspect. What are the expected consequence of a variable time-step in the comparison? 3) Equation 7 seems a Gram-Charlier series (of type A). 4) Do the authors find any issue of negative probability in their solution of the FPE using the polynomial expansion of the pdf? 5) I wonder if the solution of the SDE (1) with a much smaller time step and many more particles could be used as the reference solution, instead of the deterministic solution of the FPE. 6) It seems to me that to obtain 15 from 11 involves also the assumption that $C_{k+1}=0$. 7) Page 10, line 7. I think that (15) should be (16). 8) Page 11, line 15. May be it is worth commenting that the difference between the particles concentration and the concentration of a tracer is only in the normalization. 9) Page 18 line 6. I think it should be "we do not recommend it for ….". 10) Page 19 line 5-6. I think the phrase "…, all can obtained easily from results found in (see chapter 22…)" should be rewritten. 11) Eq. 18 and Appendix D. It would be better to use a different symbol for the bandwidth since "h" is previously used for the boundary-layer. 12) Page 25, line 6. What is $c_{zz}$?

---

## Author Comment (AC1) · 1 Jun 2016

Thank you for the encouraging and positive review. A point-by-point response follows:

1. Pg. 4. Typo. Fixed, thank you.

2. Pg. 4. Definition of $\omega$. After equation (3) we have added the sentence 'Explicitly here $\omega = w/\sigma_w$.'

3. Pg. 5. l. 16 Re: Explicit statement of initial conditions: We have added an equation at this point explicitly stating the initial conditions.

[Figure]

4. Pg. 6 Equation (16) Re: Random walk model: We have added a comment in the text after equation 16:
'Note that the RDM model can be derived formally from the LPDM in the distinguished limit of short decorrelation time, $\sigma_w \to \infty$, $\tau \to 0$ with $\sigma_w^2 \tau = \kappa$ finite (see sec. 6.3 of Rodean, 1996).

5. Pg. 9 l. 23 and Fig 5 typos.. Fixed, thanks.

6. Appendix A, integral identity. We have added the sentence and equation
'Notice that a special case of (A3), for $j = 0$, is the integral identity $\int_{-\infty}^{\infty} \mathrm{He}_k(\omega) \mathrm{e}^{-\omega^2/2} \, \mathrm{d}\omega = 0, \quad (k \geq 1)$.

7. Failure to reference Fig. 1 . Thank you for spotting this! We have added to the sentence on pg. 3. l. 29.
'The details of the profiles used are given in Table 1 and are plotted in Fig. 1.'

---

## Author Comment (AC2) · 1 Jun 2016

Thank you for the encouraging and positive review. A point-by-point response follows:

1. Calculation times for different schemes: We considered making further comment on calculation speeds in the original version, but were concerned about reproducibility due to different programming languages, machine architectures etc.. Encouraged by the referee, however, we have now added a table comparing computational clock times, measured relative to E-M, for all of the schemes considered. This is table 4 in the new version and it has the following description in the text.

'In the results below, in the interests of reproducibility, the error is presented as a function of the fixed time-step $\Delta t$ for each scheme. However, the schemes have different computational costs per time-step, which will depend on both the method of implementation of each algorithm, and on the machine used for the simulations. To give a rough idea of representative computational costs, in Table 4 the relative cost, measured with reference to the E-M scheme is shown for our calculations. Following best practice in large operational calculations (see e.g. Stohl et al. 2005), the random numbers used to simulate the Wiener processes are pre-calculated so the costs of their generation are not included in the comparison.'

2. Variable versus fixed time steps: For a comparison between schemes it seemed to us that using a fixed time step was a sensible starting point. We agree with the referee, however, that it is interesting to compare the schemes with fixed and variable time-steps. We have therefore made some variable time-step calculations (with $\Delta t \propto \tau$) and have summarised our findings in the following paragraph added to the last paragraph in section 4.1

'Another possible computational saving comes from the use of variable time-steps. To test whether or not a significant computational saving is easily attainable, we have made some calculations in which $\Delta t \propto \tau$ (the local Lagrangian decorrelation time). For each scheme tested, the use of variable time-steps was found to lead to a computational saving of a factor of around two to three compared to fixed time-steps, with the schemes otherwise performing as detailed below. More details on variable time-stepping schemes will be given elsewhere.'

3. Gram-Charlier series of Type A: We have added a comment about this on pg. 5. 'In statistics this expansion is also known the Gram-Charlier series of Type A (see pg. 23 of Barndorff-Nielsen and Cox, 1989)'.

4. Negative probability in the FPE solution: We agree that in theory a truncated series can certainly lead to negative values in the approximation to $p(\omega, z, t)$. However, we do not use $p(\omega, z, t)$ explicitly anywhere, and the only practical problem in which the very small errors (order $10^{-16}$) associated with our truncation might be important, is if one were interested in the probability of trajectories reaching a very high velocity (i.e. extreme value statistics).

5. Reference solution from a large SDE ensemble and small time-step: Yes, one could certainly construct the reference solution using the SDEs (in fact we have effectively demonstrated this in our paper), but for a given degree of accuracy this would be necessarily be much more computationally expensive compared to the FPE method we have chosen.

6. Derivation of the diffusion equation: Our derivation is to replace the $k = 1$ equation in our hierarchy (11) with the quasi-steady approximation in equation (14). This assumes that both $C_2 \approx 0$ and $\partial_t C_1 \approx 0$ and we do not justify it formally (it is just a truncation of the series). However, we have added a reference to the formal derivation of the diffusion equation from the LPDM (see reply 4 to reviewer 1).

7. Reference to equation (15)/(16): Fixed, thanks.

8. Pg. 11. Particle concentration versus concentration of tracer: After equation (6) we have added the comment in parenthesis '(In general, tracer concentrations and the marginal probability given in (6) can differ by a normalisation constant.)'

9. Pg. 18. typo. Fixed, thanks.

10. Pg. 19. l. 5-6 phrase. Fixed, thanks.

11. Repeated symbol '$h$' for bandwidth: $h$ is now changed to $h_b$ for bandwidth, and $h_*$ remains as optimal bandwidth.

**GMDD**

12. Definition of $c_{zz}$: This is simply the second derivative of the particle concentration $c(z,t)$ with respect to $z$. This symbol has now been changed to $\partial_{zz}c(z,t)$ in the text.
* * *

---

## Author Response (AR2)

**Replies to topical editor**

Thank you for the encouraging and positive comments. A point-by-point response follows:

1. Units for time $t$ and time-step $\Delta t$: Thanks for spotting this, the whole manuscript has now been checked and time units are now consistent throughout the paper.

2. Pg.11 l. 14. use of citep instead of citet: Fixed, thank you.

3. Pg.15 l. 18. phrase: Fixed, thank you.

4. Figs. 4-6 absence of EXPLICIT 3.0 long time-step results: The reason for this is due to problems with reflective boundary conditions, as explained in the last paragraph of section 4.2. In Pg. 10 l. 16, we have added the following comment referring to the EXPLICIT 3.0 at long time-steps:
   "It was not found to be possible to obtain solutions for EXPLICIT 3.0 using time-steps longer than $\Delta t = 0.02 \, h/u_*$ because of problems with the reflective boundary conditions."

**Replies to reviewers**

**Reviewer 1**

Thank you for the encouraging and positive review. A point-by-point response follows:

1. Pg. 4. Typo. Fixed, thank you.

2. Pg. 4. Definition of $\omega$. After equation (3) we have added the sentence 'Explicitly here $\omega = w/\sigma_w$.'

3. Pg. 5. l. 16 Re: Explicit statement of initial conditions: We have added an equation at this point explicitly stating the initial conditions.

4. Pg. 6 Equation (16) Re: Random walk model: We have added a comment in the text after equation 16:
'Note that the RDM model can be derived formally from the LPDM in the distinguished limit of short decorrelation time, $\sigma_w \to \infty$, $\tau \to 0$ with $\sigma_w^2 \tau = \kappa$ finite (see sec. 6.3 of Rodean, 1996).

5. Pg. 9 l 23 and Fig 5 typos.. Fixed, thanks.

6. Appendix A, integral identity. We have added the sentence and equation
'Notice that a special case of (A3), for $j = 0$, is the integral identity

$$\int_{-\infty}^{\infty} \mathrm{He}_k(\omega)\mathrm{e}^{-\omega^2/2} \, \mathrm{d}\omega = 0, \quad (k \geq 1).$$

7. Failure to reference Fig. 1 . Thank you for spotting this! We have added to the sentence on pg. 3. l. 29
'The details of the profiles used are given in Table 1 and are plotted in Fig. 1.'

**Reviewer 2**

Thank you for the encouraging and positive review. A point-by-point response follows:

1. Calculation times for different schemes: We considered making further comment on calculation speeds in the original version, but were concerned about reproducibility due to different programming languages, machine architectures etc.. Encouraged by the referee, however, we have now added a table comparing computational clock times, measured relative to E-M, for all of the schemes considered. This is table 4 in the new version and it has the following description in the text.

   'In the results below, in the interests of reproducibility, the error is presented as a function of the fixed time-step $\Delta t$ for each scheme. However, the schemes have different computational costs per time-step, which will depend on both the method of implementation of each algorithm, and on the machine used for the simulations. To give a rough idea of representative computational costs, in Table 4 the relative cost, measured with reference to the E-M scheme is shown for our calculations. Following best practice in large operational calculations (see e.g. Stohl et al. 2005), the random numbers used to simulate the Wiener processes are pre-calculated so the costs of their generation are not included in the comparison.'

2. Variable versus fixed time steps: For a comparison between schemes it seemed to us that using a fixed time step was a sensible starting point. We agree with the referee, however, that it is interesting to compare the schemes with fixed and variable time-steps. We have therefore made some variable time-step calculations (with $\Delta t \propto \tau$) and have summarised our findings in the following paragraph added to the last paragraph in section 4.1
   'Another possible computational saving comes from the use of variable time-steps. To test whether or not a significant computational saving is easily attainable, we have made some calculations in which $\Delta t \propto \tau$ (the local Lagrangian decorrelation time). For each scheme tested, the use of variable time-steps was found to lead to a computational saving of a factor of around two to three compared to fixed time-steps, with the schemes otherwise performing as detailed below. More details on variable time-stepping schemes will be given elsewhere.'

3. **Gram-Charlier series of Type A**: We have added a comment about this on pg. 5. 'In statistics this expansion is also known the Gram-Charlier series of Type A (see pg. 23 of Barndorff-Nielsen and Cox, 1989)'.

4. **Negative probability in the FPE solution**: We agree that in theory a truncated series can certainly lead to negative values in the approximation to $p(\omega, z, t)$. However, we do not use $p(\omega, z, t)$ explicitly anywhere, and the only practical problem in which the very small errors (order $10^{-16}$) associated with our truncation might be important, is if one were interested in the probability of trajectories reaching a very high velocity (i.e. extreme value statistics).

5. **Reference solution from a large SDE ensemble and small time-step**: Yes, one could certainly construct the reference solution using the SDEs (in fact we have effectively demonstrated this in our paper), but for a given degree of accuracy this would be necessarily be much more computationally expensive compared to the FPE method we have chosen.

6. **Derivation of the diffusion equation**: Our derivation is to replace the $k = 1$ equation in our hierarchy (11) with the quasi-steady approximation in equation (14). This assumes that both $C_2 \approx 0$ and $\partial_t C_1 \approx 0$ and we do not justify it formally (it is just a truncation of the series). However, we have added a reference to the formal derivation of the diffusion equation from the LPDM (see reply 4 to reviewer 1).

7. **Reference to equation (15)/(16)**: Fixed, thanks.

8. **Pg. 11. Particle concentration versus concentration of tracer**: After equation (6) we have added the comment in parenthesis '(In general, tracer concentrations and the marginal probability given in (6) can differ by a normalisation constant.)'

9. **Pg. 18. typo.** Fixed, thanks.

10. **Pg. 19. l. 5-6 phrase.** Fixed, thanks.

11. **Repeated symbol '$h$' for bandwidth:** $h$ is now changed to $h_b$ for bandwidth, and $h_*$ remains as optimal bandwidth.

12. **Definition of $c_{zz}$**: This is simply the second derivative of the particle concentration $c(z, t)$ with respect to $z$. This symbol has now been changed to $\partial_{zz} c(z, t)$ in the text.

[revised manuscript text omitted]